# MICFuzzy: A maximal information content based fuzzy approach for reconstructing genetic networks

**Hasini Nakulugamuwa Gamage**[1]*, **Madhu Chetty**[1]*, **Suryani Lim**[1], **Jennifer Hallinan**[2]

**1** Health Innovation and Transformation Centre, Federation University, Churchill, Victoria, Australia,
**2** BioThink, Queensland, Australia

* hasininakulugamuwagamage@students.federation.edu.au (HNG); madhu.chetty@federation.edu.au (MC)

## Abstract

In systems biology, the accurate reconstruction of Gene Regulatory Networks (GRNs) is crucial since these networks can facilitate the solving of complex biological problems. Amongst the plethora of methods available for GRN reconstruction, information theory and fuzzy concepts-based methods have abiding popularity. However, most of these methods are not only complex, incurring a high computational burden, but they may also produce a high number of false positives, leading to inaccurate inferred networks. In this paper, we propose a novel hybrid fuzzy GRN inference model called MICFuzzy which involves the aggregation of the effects of Maximal Information Coefficient (MIC). This model has an information theory-based pre-processing stage, the output of which is applied as an input to the novel fuzzy model. In this preprocessing stage, the MIC component filters relevant genes for each target gene to significantly reduce the computational burden of the fuzzy model when selecting the regulatory genes from these filtered gene lists. The novel fuzzy model uses the regulatory effect of the identified activator-repressor gene pairs to determine target gene expression levels. This approach facilitates accurate network inference by generating a high number of true regulatory interactions while significantly reducing false regulatory predictions. The performance of MICFuzzy was evaluated using DREAM3 and DREAM4 challenge data, and the SOS real gene expression dataset. MICFuzzy outperformed the other state-of-the-art methods in terms of F-score, Matthews Correlation Coefficient, Structural Accuracy, and SS_mean, and outperformed most of them in terms of efficiency. MICFuzzy also had improved efficiency compared with the classical fuzzy model since the design of MICFuzzy leads to a reduction in combinatorial computation.

## Introduction

A major challenge in systems biology is the accurate inference of gene regulatory networks (GRNs), which are used for investigating a range of complex biological processes. The development of microarray and sequencing technologies has produced large amounts of gene expression data for the reconstruction and analysis of GRNs. During gene expression, target genes

**Funding:** HG - receiving a tuition fee waiver scholarship from Federation University Australia and a Destination Australia stipend scholarship from the Australian Government Department of Education, Skills and Employment for pursuing this doctoral research - The funders had no role in study design, data collection and analysis, decision to publish, or preparation of the manuscript.

**Competing interests:** The authors have declared that no competing interests exist.

are controlled by their regulators [1] and these regulatory interactions are inferred using a plethora of reverse engineering approaches. Among these approaches, Boolean modelling, a simple and efficient modelling approach, can capture both the structure and dynamics of a regulatory network [2, 3]. However, these models require the use of a data discretization method which leads to information loss. Due to this requirement, the results produced by most of the Boolean Network models for GRN inference are near-optimal [2, 3].

Bayesian network modelling is more sophisticated than Boolean modelling and is used to implement high performing reverse engineering approaches [4–8]. Perrin's method, based on dynamic Bayesian networks is well-suited to the inference of gene regulatory interactions from gene expression data and their derivatives. However, this approach is limited to the inference of small-scale GRNs [4]. Bayesian Network inference with Java Objects (Banjo), is a software package that has been used for both Bayesian and Dynamic Bayesian network inference. Similar to Perrin's approach, Banjo requires a high computational time, since it uses heuristic search strategies in model learning [5]. Morshed et al. [6] implemented a Bayesian network model to capture both instantaneous and time-delayed interactions that occur concurrently, but the evolutionary search employed requires high computational time [6]. Unlike other Bayesian models, GlobalMIT uses mutual information test (MIT), an information theoretic-based scoring metric as a model learning technique rather than time-consuming local search strategies. However, this approach is poorly scalable since the complex nature of Bayesian modelling makes it computationally intensive when inferring large-scale networks [7].

Differential equations (DE) [28], are a deterministic modeling technique used in Computational Biology that has the ability to model the dynamic behavior of GRNs. These models incur a high computational burden when learning model parameters with the use of optimization techniques, and are not suited to the reconstruction of large-scale networks [9–11]. Machine learning-based methods such as regression-based approaches have been used for efficient and accurate network inference [12, 13]. Gene networks inference using projection and lagged regression (GNIPLR) [14] uses both projection and lagged regression strategies to accurately infer GRNs from time series and non-time series data. LassoCV+RidgeCV [12] is another regression-based model which uses an improved version of both regression methods, incorporating cross-validation to increase model stability and produce accurate results. Both GNIPLR and LassoCV+RidgeCV outperform other existing high-performing regression-based inference methods. However, these regression methods are limited to capturing linear dependencies.

Given pros and cons of the above-mentioned approaches, information theory-based methods have gained increasing attention in recent years [3, 15, 16] because of their ability to effectively capture associations between genes from high dimensional gene expression data with limited sample size [17]. Foremost among these methods are those using Mutual Information (MI) to quantify not only linear dependencies between the target and the regulatory genes, but also complex non-linear dependencies [16, 18]. Context likelihood of relatedness (CLR) [19] and MRNET [20] are existing information theory-based methods that use MI to determine transcriptional regulatory interactions. CLR produces accurate GRNs by eliminating false interactions using MI score-based background correction. MRNET uses a pair-wise mutual information evaluation method, Min-Redundancy and Max-Relevance for inferring regulatory interactions. NARROMI [21] and CMI2NI [22] are other novel MI-based methods for GRN inference. NARROMI, a combination of the ordinary differential equation-based recursive optimization (RO) method and the information theory-based mutual information (MI) method, is an effective method which outperforms most existing methods in terms of accuracy and false positive rates. In this approach, the least relevant regulators for each target are first removed using MI by evaluating pairwise correlations. Then indirect regulators are identified

using recursive optimization, which further improved the overall model accuracy [21]. CMI2NI uses a novel association measure, conditional mutual inclusive information (CMI2) which helps to identify direct regulations while eliminating indirect regulations. The main drawback of CMI2 is that its efficiency is negatively impacted by the use of a random method to identify conditional genes [22].

Some recently developed methods combine information theoretic concepts with other techniques to eliminate model specific issues. KFGRNI is such a method, using Conditional Mutual Information (CMI) based approach to fine-tune a list of genes, selected by the ensemble Kalman filter and regression approach [23]. This method further improves model accuracy by removing false regulations. KFLR [24] uses MI and CMI in the preprocessing stage to eliminate noisy regulations, followed by a Kalman filter-based model averaging approach (a hybrid framework of Bayesian model averaging and linear regression methods) to infer possible regulators. However, both MI and CMI-based inferencing methods cannot discover important non-linear correlations such as sinusoids [15, 21]. MI is well suited for use on discrete or categorical data [18], but has known limitations when applied to continuous data, as is found in gene expression datasets.

To overcome these limitations of MI, the Maximal Information Coefficient (MIC) which has been designated "correlation for the 21st century" [25, 26], was introduced to capture functional and non-functional associations between continuous variables. MIC has produced impressive performances by detecting strengths and patterns of dependencies in high-dimensional datasets with limited sample size, making it a promising tool for GRN inference [27]. MIC is also known to be able to identify a wide range of interesting associations, making it valuable for the evaluation of the regulatory associations between genes. Compared to other available correlation measures, which are limited in their application to discretized data, MIC is well suited to continuous data, such as microarray gene expression data [18]. As demonstrated in [28], MIC is also noise resistant which makes it well suited to GRN inference since real gene expression data usually contains high amounts of noise. Therefore, recently developed GRN inference methods (i.e., MRNETB) [29, 30] have incorporated MIC to improve the performance of existing methods such as CLR and MRNET. Recently MICRAT has been developed, using MIC to infer GRNs as an undirected graph that represents interactions between genes from time series gene expression data. Then the direction of these interactions is determined using a combination of conditional relative average entropy and time course mutual information of pairs of genes [15]. However, these models produce a high number of false regulatory predictions while inferring a high number of true regulations and have no ability to identify the activating or inhibiting effect of regulatory genes.

Due to their ability to deal with uncertain and imprecise information fuzzy concept-based models for GRN inference [31–36] have been extensively investigated in the last decade [36]. Fuzzy logic, a knowledge-based problem-solving approach, consists of three main stages: fuzzification, inference, and defuzzification. When applying fuzzy logic to GRN inference, a typical classical fuzzy model reported by Woolf and Wang [31] has been implemented. In this approach, the expression levels of activator and repressor genes are mapped to the set of linguistic values High, Medium, and Low, based on fuzzy membership functions. In the inference stage, the paired activator and repressor genes are used to predict the expression value of a target gene using fuzzy rules after applying these pairs to the decision matrix. At points where their predictions overlap, a fuzzy score for the predicted target variable is generated [31]. These generated scores from each GRN fuzzy rule are aggregated using an accumulation method to obtain the final fuzzy value [36, 37]. In defuzzification, these fuzzy predicted gene expression values are transformed back to crisp gene expression values based on the membership functions of the target gene. Either the centroid defuzzification or the center of gravity (COG) method is used as a

defuzzification technique in the GRN inference context [36, 38]. Later implementations of classical fuzzy models included several enhancements of each step [36].

Variations of classical fuzzy models, incorporating changes in gene expression levels, activator-repression interactions, or pre-processing methods, have also been reported [32, 36, 39]. In the rest of this paper, it may be noted that any reference to the "classical model" implies the initial version of the classical fuzzy model reported by Woolf and Wang [31]. The main drawback of all classical fuzzy models, compared to other methods, is their high computational burden. Hybrid fuzzy models combining classical fuzzy models with other methods, have been found to be effective in improving the performance of classical fuzzy models [36]. A hybrid fuzzy model introduced by Sokhansanj and Fitch [40], applied Union Rule Configuration (URC) together with fuzzy logic to avoid the combinatorial explosion issue in fuzzy rules. Another hybrid model by Poblete and colleagues [41], combines fuzzy inference systems with ODEs to produce an accurate model by reducing the need of prior knowledge. However, this approach is limited to the inference of small-scale networks. In addition to classical fuzzy models, fuzzy logic-based tools have been introduced, such as fuzzy cognitive maps (FCM), a powerful intelligent tool for GRN inference. LASSO-FCM, a least absolute shrinkage and selection operator-based FCM [33], compressed sensing-based FCM (CF-FCM) [34], and Kalman filter and compressed sensing-based FCM (KFCSFCM) [35] are recently developed accurate methods. However, these methods have known limitations in model learning [36].

In this research, we developed a novel hybrid fuzzy model, which combines a novel fuzzy model with the key feature of the information theory-based approach. This hybrid fuzzy model, called MICFuzzy, exploits the effectiveness of the MIC [42] for pre-processing and as a measure of the strengths of regulatory relationships between genes. To the best of our knowledge, the combination of MIC measure with a fuzzy GRN model is novel, and has not been investigated earlier. In MICFuzzy, MIC is used to evaluate gene dependencies and identify the most informative regulatory genes for each target gene, while eliminating genes with a lower probability of being a regulatory gene from the high-dimensional gene space. This pre-processing reduces the computational burden by pruning the possible number of activator and repressor gene pairs to be considered in the fuzzy model. In the fuzzy segment of the hybrid model, the regulatory genes for each target gene are identified from the list of filtered genes by inferring the activator-repressor gene pairs which can most closely approximate the target gene expression level. Rather than considering only the effect of the input gene expression values of the activator and repressor pairs when predicting the target gene expression levels, in the proposed model, a novel metric called regulatory relationship strength (RRS) is used. Unlike other classical fuzzy models, which are commonly limited to selecting only one activator and repressor gene pair, the MICFuzzy model allows the selection of more than one suitable gene pair. This model is also capable of identifying the regulatory relationship type (activator or repressor) between genes. The results of the MICFuzzy model obtained using the DREAM3 and DREAM4 datasets [43–45], along with the real word SOS gene repair dataset [46], demonstrated its superiority compared to other state-of-the-art methods.

The rest of the paper is organized as follows. Section 2 deals with the methodology and the steps used in building our proposed hybrid model. In Section 3, we discuss the results obtained from the experiment and the comparison of the performance of our model with other state-of-the-art methods. Finally, we conclude the paper in Section 4.

## Methods

This section presents a concise overview of our proposed model, MICFuzzy (Fig 1), a novel framework for GRN inference. In MICFuzzy, the GRN inference problem is first divided into

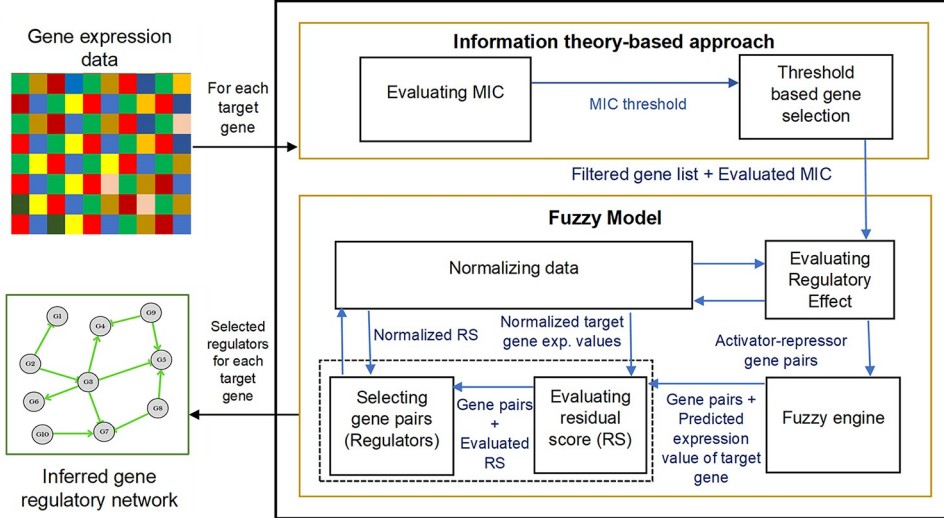

**Fig 1. Schematic diagram of the proposed hybrid fuzzy model.** The proposed model introduces two main components: an information theory-based approach and a fuzzy model. In the fuzzy model, the dashed box indicates the steps by which the regulatory genes are selected.

*n* sub-problems, where *n* is the number of genes in the network. The proposed model is applied to each sub-problem to identify the potential regulators for each target gene, using time series gene expression data. The MICFuzzy model has two main stages. In the pre-processing stage, an information theory-based approach identifies the list of the most promising regulators for each target gene from the high-dimensional gene space. Then, the novel fuzzy model selects the best regulator gene list for each target gene, reducing false predictions while accurately identifying activator and repressor regulatory genes. The pseudo-code for the MIC-Fuzzy algorithm is given in S1 Fig.

## Information theory-based approach: The pre-processing stage

An information theory-based approach is used to filter the relevant genes for each target gene. This stage helps to reduce the computational burden of the fuzzy model used in the next stage. In this stage, we first evaluate the pair-wise MIC of every other gene corresponding to the target gene under consideration. The MIC is a metric which can capture simple to complex dependencies in both functional and non-functional relationships. It has two properties: generality and approximate equitability. Generality permits the detection of both functional and non-functional relationships from data with a sufficient sample size, while equitability gives similar scores for equally noisy relationships, regardless of the type of the relationship [25]. Since MIC has the demonstrated ability to deal with the noise in gene expression data it is highly suitable to use as a preprocessing method [28]. The maximal information coefficient of a set of two-variable (*X* and *Y*) data pairs, *D* can be defined as follows:

$$MIC(D) = \max_{XY < B(n)} \left\{ \frac{I^*(D, X, Y)}{\log_2(\min(X, Y))} \right\} \tag{1}$$

where, *D* consists of pair of values $<x,y>$, $x \in X$, $y \in Y$ in a sample of size *n*. The grid size is $XY < B(n)$, where the function of sample size, $B(n) = n^{0.6}$ is the default setting to achieve high performance. The term, $I^*(D, x, y)$ is the maximum mutual information taken over an *x* and *y* pair. We use $\log_2(\min(x, y))$ to normalize the maximum mutual information [15, 25]. MIC has

been reported [25, 29] to perform effectively in measuring the strength of association in two-variable relationships.

Based on Eq (1), in GRN inference, $X$ is the candidate gene and $Y$ is the target gene. When evaluating the MIC between each gene pair, the gene expression value of $X$ ($x$) at time point $t$ and gene expression value of $Y(y)$ at time point $t+1$ is considered. Then, based on a user-defined threshold MIC value, highly correlated genes are identified as the selected gene set for each target gene. The model allows the user to decide the MIC threshold empirically in a specific problem context. In this study, the threshold value was computed as the average of the MIC values corresponding to a target gene, to reduce the experimental burden. In the next stage, the selected gene list and the evaluated MIC scores are input to the novel fuzzy model to further enhance the accuracy of the selection of the regulatory genes.

## Novel fuzzy model

We improved upon the classical model [31] by considering the regulatory effect, Eq (2), to determine the expression level of the target gene. The regulatory effect consists of two factors: the gene expression levels of regulatory genes (activator-repressor pairs) and the strength of the regulatory relationship between an activator or repressor gene and the target gene:

$$\textit{Regulatory effect} = \textit{Regulatory Relationship Strength} * \textit{Gene expression level} \qquad (2)$$

In the classical fuzzy model, only the expression levels of regulatory genes are considered when predicting the expression level of the target gene [36]. In our study, the MIC score of a gene corresponding to a target gene is considered, to reflect the strength of the regulatory relationship. The evaluated regulatory effect is normalized using min-max normalization, with the values scaled between 0 and 1. This normalized regulatory effect is fuzzified into three discrete levels, Low, Medium, or High (Fig 2). For example, if the normalized regulatory effect of an activator or repressor is 0.4, then the fuzzified membership values will be Low = 0.2, Medium = 0.8, and High = 0 (Fig 2).

After forming activator-repressor pairs from the filtered gene set, using the set of heuristic fuzzy rules that we have defined based on the existing expert knowledge and the activator/repressor regulatory rule described in [31, 47], the expression level of the target gene is predicted. The target gene expression has five different discrete levels (Fig 3). For example, referring to Fig 4, we have a fuzzy rule:

*"If an activator's regulatory effect is **High** AND the repressor's regulatory effect is **Low** then the target gene expression level will be **Very High**"*

The discrete levels for both input and output variables are decided based on previously reported research [47] which demonstrated these discrete levels are more accurate than those in classical model. In this step, the bounded sum technique is used to combine the results from each fuzzy rule [37]. Finally, the fuzzified output obtained for the target gene is defuzzied back to a crisp value using the centroid defuzzification method. This process is continued until the target gene expression level has been estimated for each activator-repressor gene pair for all available timepoints in the dataset.

**Selection of regulatory genes.** After the predicted gene expression value of a target gene for each time point is obtained, this predicted value is compared with the actual expression value of the target gene for each time point to evaluate the Mean Squared Error (MSE) as

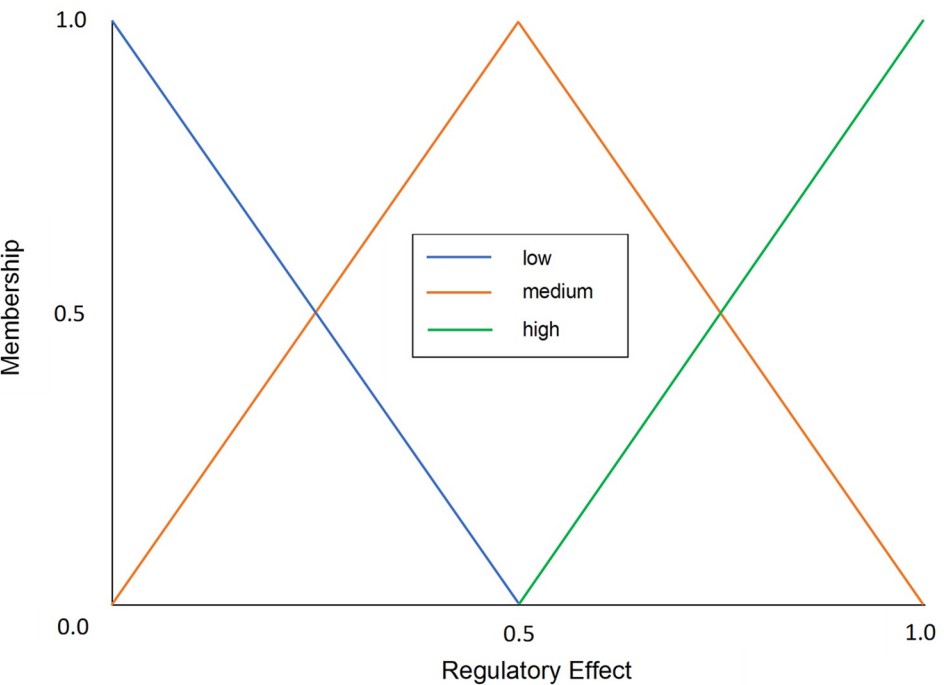

**Fig 2. Fuzzy membership function of the regulatory effect of repressors and activators.** It has three discrete levels: Low, Medium, and High.

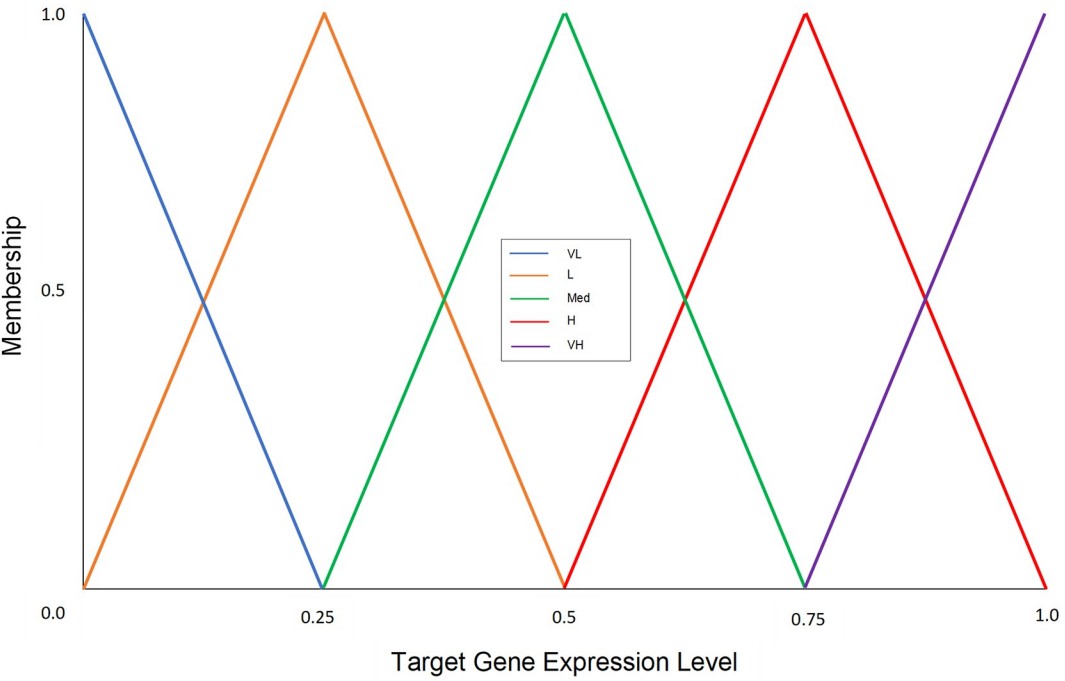

**Fig 3. Fuzzy membership function of target gene expression level.** There are six discrete levels: VL = Very Low, L = Low, Med = Medium, H = High, and VH = Very High.

## If Repressor (Regulatory Effect) is

**Fig 4. Fuzzy decision rule matrix to predict the expression level of a target gene based on the regulatory effect of activator and repressor gene pairs.**

follows:

$$MSE = \frac{1}{T}\sum\nolimits_{t=2}^{T}\left(Y_t - \hat{Y}_t\right)^2 \tag{3}$$

where $T$ is the total number of observations (time points) in the dataset, and $Y_t$ and $\hat{Y}_t$ are the experimental and the corresponding predicted target gene expression values, respectively.

A second performance metric, variance, has been used to measure the acceptability of the predicted results. Statistical variance is measured based on the number of rules fired by a triplet when finding the target gene expression level. If all fuzzy rules are evenly fired by a gene triplet over all time points, then we consider the variance of this triplet to be low, and the predicted target gene expression level from the considered repressor and activator pair to be acceptable. Based on these two metrics, the final residual score (RS) is evaluated as follows:

$$Residual\ Score = MSE * Variance \tag{4}$$

The best set of activator-repressor pairs for each target gene is selected based on the RS value. To be the best pair, the RS should be the lowest. The evaluated RS scores of activator-repressor pairs for a candidate target gene are normalized to a value between 0 and 1 (*NRS* in Algorithm in S1 Fig) to decide the common threshold level of RS for each target gene. Here, we have assumed that one gene can act as either a repressor or an activator for a candidate

target gene in a given problem context [10]. Finally, based on the RS threshold thus defined, the genes in the activator-repressor pairs with lower RS than the defined threshold are selected as the best set of regulatory genes for a target gene (Algorithm in S1 Fig). After selecting a suitable regulatory gene list for each target gene (each sub-problem) the overall network structure can be obtained.

## Results and discussion

To evaluate the proposed model, experiments were first carried out on the simulated datasets of small and medium-scale networks used in the DREAM3 and DREAM4 *in silico* challenges for gene network inference [43–45]. We used the 10- and 50-gene networks from the DREAM3 challenge. From the DREAM4 challenge, 10- and 100-gene network sub-challenges were addressed. For each of these sub-challenges, there are five different networks, referred to here as Net1, Net2, Net3, Net4, and Net5. In both the DREAM3 and DREAM4 challenges there are 21 time points for each time series. However, the majority of existing fuzzy-based reverse engineering approaches consider only the last 11 points in each time series, from which the perturbation has been removed [33, 34]. We considered all time points in each network in each sub-challenge when inferring the GRNs, including the effect of perturbations. The proposed approach was also studied using a real gene expression dataset, the well-studied SOS DNA repair network of *Escherichia coli*. This network contains eight genes: uvrD, lexA, umuD, recA, uvrA, uvrY, ruvA, and polB. The SOS DNA dataset was obtained from four different experiments under various UV light conditions, with the gene expression measured at 50 time points evenly spaced at six-minute intervals [46].

We evaluated the performance of MICFuzzy under different conditions, with and without the effect of regulatory relationship strength, and compared the results with the individual performance of the information theory-based approach (pre-processing stage). The accuracy of network inference using MICFuzzy was also compared with the state-of-the-art methods, MICRAT, NARROMI, CMI2NI, LASSO-FCM, CF-FCM, and KFCSFCM, and the improvement in efficiency was mainly compared with the classical model. The performance of MICFuzzy on the SOS dataset was compared with that of MICRAT and other well-known state-of-the-art methods [15].

The inferred network topology was evaluated using popular performance metrics, including Precision, True Positive Rate/Recall/Sensitivity, False Positive Rate, Specificity, F-score, SS_mean, Matthews correlation coefficient (MCC), and Structural Accuracy. F-score, Structural Accuracy, and MCC are the most reliable measures, as demonstrated in [15, 48], which produce high values if there is a balance in producing good results in all four categories (True Positive, False Positive, True Negative, and False Negative). Therefore, in this research, we used F-score, Structural Accuracy, and MCC as performance evaluation measurements. In addition, SS_mean, a sensitivity specificity-based measure, was also used, since the results of most of the fuzzy concepts-based models for GRN inference are reported in terms of SS_mean [33–35, 49].

### Experimental setup and parameter settings

MICFuzzy, implemented in Python, uses the *minepy* Python library [50] to evaluate the MIC between each gene pair in the pre-processing stage. The *minepy* library has identical performance on biological datasets as its original java implementation (*mine.jar*) [51] using the default settings published in [25]. We also applied the *Scikit-fuzzy* library [52], to build the fuzzy control system. MICFuzzy also utilised *NumPy*, *SciPy*, *pandas*, *statistics*, and *Matplotlib*

scientific libraries. The steps of the implementation of MICFuzzy and its libraries are explained in S1 Fig.

The hybrid fuzzy model consists of several parameters which must be fine-tuned to obtain high structural accuracy: a high number of true regulations and a low number of false predictions. In the first stage of the hybrid model, we need to define the MIC threshold, while in the fuzzy model, the normalized RS threshold must be defined. To reduce the experimental burden, we used the average of measured values for both the MIC threshold and the RS threshold. Thus, for the three 10-gene networks, Net1, Net4, and Net5 of DREAM3, the MIC threshold was set to 0.3, while for Net2 the threshold was set to 0.29, and for Net3 to 0.25. The normalized RS thresholds for Net1, Net2, Net3, Net4, and Net5 were set to 0.03, 0.09, 0.03, 0.02, and 0.03, respectively. For the 50-gene network, the MIC threshold was set to 0.4 for all five networks, and rather than using a RS threshold, in this context we selected the first 10 activator and repressor pairs with low normalized RS values. For the 10-gene network from the DREAM4 challenge, the MIC thresholds for Net1 to Net5 were set to 0.35, 0.39, 0.3, 0.3, and 0.4, respectively, while the normalized RS thresholds were set to 0.05, 0.35, 0.09, 0.13, and 0.05, respectively. For the 100-gene network, the MIC threshold was set to 0.5 for all five networks, and rather than using a RS threshold, in this context we selected the first 20 activator and repressor pairs with low normalized RS values. When inferring the SOS DNA network, the MIC threshold, and RS threshold were set to 0.37 and 0.18 respectively.

## Experiments on DREAM3 and DREAM4 datasets

Experiments on the performance of the information theory-based approach identified a high number of true regulations while producing a relatively high number of false positives (Table 1) in all network inference problems. For example, for the DREAM3 10-gene Net2, and the DREAM4 10-gene Net5 inference problems, the information theory-based approach identified all the true regulations, giving a true positive rate of 1.0, while producing false positive rates of 0.48 and 0.51, respectively (Table 1). These results indicate that, as a preprocessing method, the information theory-based approach contributed to improving the overall performance of the MICFuzzy by inferring a high number of true positives at the initial stage.

**Integration of regulatory relationship strength.** The overall improvement in performance produced by the MICFuzzy model was tested under two conditions: with and without the effect of regulatory relationship strength. The results, shown in Table 1 show that in 10-gene, 50-gene and 100-gene inference problems the proposed fuzzy model considerably reduced the number of false positives inferred by the information theory-based approach. However, without the effect of the regulatory relationship strength, this reduction was also accompanied by a detrimental reduction in the number of true regulations identified. By incorporating the regulatory relationship strength into the regulatory effect for the fuzzy model, we observed a further reduction in the false regulatory predictions, and the results retained a considerable number of identified important true regulations from the pre-processing stage, prioritizing highly correlated genes for the 10-gene, 50-gene and 100-gene networks (Table 1). In the 10-gene Net2 inference problem of the DREAM4 challenge, MICFuzzy, in both conditions, with and without the inclusion of regulatory relationship strength, performed similarly. These results occurred because each gene in the filtered gene set from the information theory-based approach had a similar regulatory relationship strength (MIC value), and the target gene expression was highly dependent on input gene expression levels. However, in all of the other networks of both the DREAM3 and DREAM4 challenges, the MICFuzzy incorporating regulatory relationship strength outperformed the model without regulatory relationship strength in terms of F-score, MCC, and Structural Accuracy (Table 1). This result further

**Table 1. Comparison of True Positive Rate, False Positive Rate, F-score, Matthews correlation coefficient, and Structural Accuracy produced by the Information theory-based approach, and MICFuzzy with and without the inclusion of regulatory relationship strength.**

| Dataset | | Method | TPR | FPR | F-score | MCC | Structural Accuracy |
|---|---|---|---|---|---|---|---|
| DREAM3 10-Gene | Net 1 | Information theory-based approach | **0.81** | 0.39 | 0.35 | 0.28 | 0.63 |
| | | MICFuzzy (without RRS) | 0.36 | 0.24 | 0.24 | 0.09 | 0.71 |
| | | MICFuzzy (with RRS) | 0.64 | **0.23** | **0.39** | **0.30** | **0.76** |
| | Net 2 | Information theory-based approach | **1.00** | 0.48 | 0.46 | **0.39** | 0.60 |
| | | MICFuzzy (without RRS) | 0.40 | 0.24 | 0.31 | 0.13 | 0.70 |
| | | MICFuzzy (with RRS) | 0.60 | **0.16** | **0.50** | **0.39** | **0.80** |
| | Net 3 | Information theory-based approach | **0.60** | 0.41 | 0.25 | 0.12 | 0.59 |
| | | MICFuzzy (without RRS) | 0.40 | 0.19 | 0.28 | 0.16 | 0.76 |
| | | MICFuzzy (with RRS) | **0.60** | **0.12** | **0.45** | **0.38** | **0.85** |
| | Net 4 | Information theory-based approach | **0.88** | 0.74 | 0.46 | 0.15 | 0.43 |
| | | MICFuzzy (without RRS) | 0.40 | 0.39 | 0.33 | 0.01 | 0.56 |
| | | MICFuzzy (with RRS) | 0.64 | **0.28** | **0.54** | **0.34** | **0.70** |
| | Net 5 | Information theory-based approach | **0.77** | 0.56 | 0.44 | 0.19 | 0.52 |
| | | MICFuzzy (without RRS) | 0.50 | 0.25 | 0.44 | 0.23 | 0.68 |
| | | MICFuzzy (with RRS) | 0.55 | **0.18** | **0.52** | **0.36** | **0.76** |
| DREAM3 50-Gene | Net 1 | Information theory-based approach | **0.45** | 0.15 | 0.12 | 0.13 | 0.84 |
| | | MICFuzzy (without RRS) | 0.24 | **0.04** | 0.19 | 0.17 | 0.94 |
| | | MICFuzzy (with RRS) | 0.39 | **0.04** | **0.28** | **0.27** | **0.95** |
| | Net 2 | Information theory-based approach | **0.40** | 0.08 | 0.22 | 0.21 | 0.91 |
| | | MICFuzzy (without RRS) | 0.24 | 0.07 | 0.15 | 0.12 | 0.91 |
| | | MICFuzzy (with RRS) | 0.35 | **0.05** | **0.26** | **0.24** | **0.93** |
| | Net 3 | Information theory-based approach | **0.37** | 0.12 | 0.14 | 0.13 | 0.86 |
| | | MICFuzzy (without RRS) | 0.21 | **0.03** | 0.21 | 0.19 | 0.95 |
| | | MICFuzzy (with RRS) | **0.37** | **0.03** | **0.33** | **0.31** | **0.96** |
| | Net 4 | Information theory-based approach | **0.45** | 0.25 | 0.18 | 0.12 | 0.73 |
| | | MICFuzzy (without RRS) | 0.30 | 0.10 | 0.22 | 0.16 | 0.86 |
| | | MICFuzzy (with RRS) | 0.39 | **0.09** | **0.30** | **0.24** | **0.88** |
| | Net 5 | Information theory-based approach | **0.45** | 0.28 | 0.18 | 0.10 | 0.71 |
| | | MICFuzzy (without RRS) | 0.33 | 0.12 | 0.23 | 0.16 | 0.84 |
| | | MICFuzzy (with RRS) | 0.44 | **0.11** | **0.32** | **0.26** | **0.86** |
| DREAM4 10-Gene | Net 1 | Information theory-based approach | **0.67** | 0.25 | 0.46 | 0.33 | 0.73 |
| | | MICFuzzy (without RRS) | 0.33 | 0.13 | 0.33 | 0.20 | 0.78 |
| | | MICFuzzy (with RRS) | 0.53 | **0.08** | **0.55** | **0.47** | **0.86** |
| | Net 2 | Information theory-based approach | **0.50** | 0.19 | 0.42 | 0.28 | 0.76 |
| | | MICFuzzy (without RRS) | **0.50** | 0.12 | 0.49 | 0.37 | 0.81 |
| | | MICFuzzy (with RRS) | **0.50** | **0.11** | **0.50** | **0.39** | **0.82** |
| | Net 3 | Information theory-based approach | **0.80** | 0.59 | 0.34 | 0.16 | 0.48 |
| | | MICFuzzy (without RRS) | 0.27 | 0.12 | 0.29 | 0.16 | 0.78 |
| | | MICFuzzy (with RRS) | 0.47 | **0.05** | **0.54** | **0.47** | **0.87** |
| | Net 4 | Information theory-based approach | **0.85** | 0.43 | 0.39 | 0.29 | 0.61 |
| | | MICFuzzy (without RRS) | 0.46 | 0.10 | 0.44 | 0.35 | 0.83 |
| | | MICFuzzy (with RRS) | **0.85** | **0.09** | **0.71** | **0.66** | **0.90** |
| | Net 5 | Information theory-based approach | **1.00** | 0.51 | 0.38 | 0.34 | 0.56 |
| | | MICFuzzy (without RRS) | 0.42 | **0.09** | 0.42 | 0.33 | 0.84 |
| | | MICFuzzy (with RRS) | 0.67 | **0.09** | **0.59** | **0.53** | **0.88** |

(*Continued*)

**Table 1.** (Continued)

| Dataset | | Method | TPR | FPR | F-score | MCC | Structural Accuracy |
|---|---|---|---|---|---|---|---|
| DREAM4 100-Gene | Net 1 | Information theory-based approach | **0.65** | 0.30 | 0.07 | 0.10 | 0.70 |
| | | MICFuzzy (without RRS) | 0.38 | 0.06 | 0.16 | 0.17 | 0.93 |
| | | MICFuzzy (with RRS) | 0.56 | **0.05** | **0.26** | **0.28** | **0.95** |
| | Net 2 | Information theory-based approach | **0.66** | 0.34 | 0.09 | 0.11 | 0.66 |
| | | MICFuzzy (without RRS) | 0.43 | 0.09 | 0.17 | 0.18 | 0.90 |
| | | MICFuzzy (with RRS) | 0.64 | **0.04** | **0.31** | **0.33** | **0.96** |
| | Net 3 | Information theory-based approach | **0.63** | 0.36 | 0.07 | 0.08 | 0.64 |
| | | MICFuzzy (without RRS) | 0.49 | 0.09 | 0.15 | 0.17 | 0.90 |
| | | MICFuzzy (with RRS) | 0.62 | **0.05** | **0.31** | **0.33** | **0.95** |
| | Net 4 | Information theory-based approach | **0.70** | 0.40 | 0.07 | 0.09 | 0.61 |
| | | MICFuzzy (without RRS) | 0.40 | 0.08 | 0.15 | 0.15 | 0.90 |
| | | MICFuzzy (with RRS) | 0.62 | **0.05** | **0.31** | **0.33** | **0.95** |
| | Net 5 | Information theory-based approach | **0.80** | 0.40 | 0.07 | 0.11 | 0.60 |
| | | MICFuzzy (without RRS) | 0.52 | 0.09 | 0.17 | 0.19 | 0.91 |
| | | MICFuzzy (with RRS) | 0.70 | **0.06** | **0.30** | **0.34** | **0.95** |

The networks considered were the 10- and 50-gene networks of DREAM3 and the 10- and 100-gene networks of DREAM4 challenge.

[a]RRS: regulatory relationship strength.

demonstrated the ability of MICFuzzy to obtain a high number of true predictions while reducing the number of false regulatory predictions.

**Comparison of MICFuzzy performance with other existing models.** Since MICFuzzy is based on concepts from both information theory and fuzzy logic, a comparison of our model was carried out on models based on information theory (CMI2NI, MICRAT and NARROMI), and with models based on fuzzy concepts (LASSO-FCM, CF-FCM, and KFCSFCM). We evaluated our model based on Average F-score, Average MCC, and Average Structural Accuracy for all five networks in the DREAM4 10-gene and 100-gene datasets, since MICRAT has used these measures for performance evaluation and thus, direct comparison is possible. As shown in Figs 5 and 6, for Average Structural Accuracy, while MICFuzzy showed a slight improvement over MICRAT (Figs 5C and 6C), it showed considerable improvement over NARROMI in 10-gene and 100-gene datasets. Our proposed model outperformed MICRAT and NARROMI substantially in both 10-gene and 100-gene datasets with respect to Average F-score (Figs 5A and 6A) and Average MCC (Figs 5B and 6B) because of its ability to identify a high number of true regulations and reduce the number of false predictions.

Compared with fuzzy concepts-based models, with respect to SS_mean, our model outperformed LASSO-FCM, CF-FCM, and KFCSFCM in inferring all five 10-gene networks in both challenges and 100-gene networks in the DREAM4 challenge (Table 2). In inferring the DREAM3 50-gene Net3, MICFuzzy outperformed other methods and was on par with KFCSFCM due to a similar SS_mean of 0.54. However, for the inference of the Net1, Net2, Net4, and Net5 networks, the MICFuzzy model outperformed CMI2NI, LASSO-FCM, CF-FCM, and KFCSFCM. Although the CMI2NI method, as an information theory based method, produced a higher performance than FCM-based models in the DREAM3 10-gene Net1, Net4, and the DREAM4 10-gene Net1, Net5 network inference its performance was inferior compared to that of the proposed model. In terms of SS_mean, while both the LASSO-FCM and CF-FCM models are accurate and recent FCM models [36], MICFuzzy produced superior performance (Table 2).

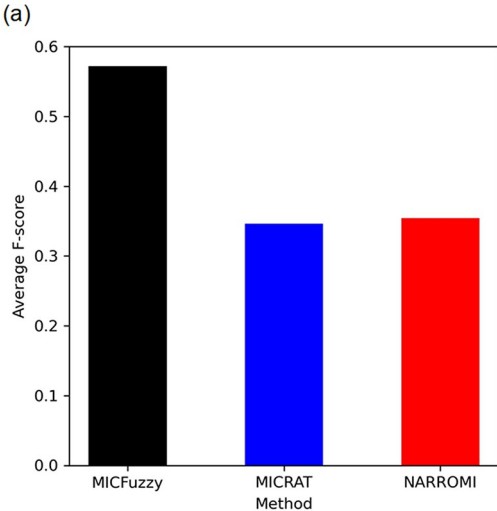

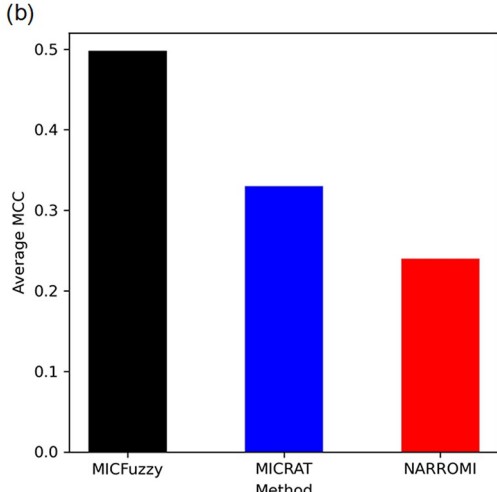

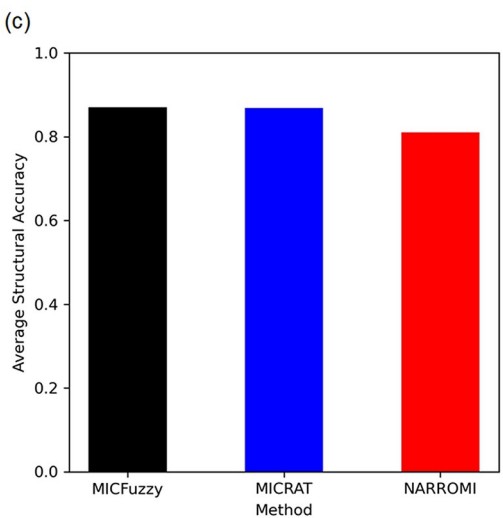

**Fig 5. Comparison of Average F-score, Average MCC and Average Structural Accuracy of MICRAT, NARROMI, and MICFuzzy for inferring DREAM4 10-gene networks.** Results of (a) Average F-score (b) Average MCC and (c) Average Structural Accuracy.

**Impact of pre-processing on computational time.** The main drawback of the classical model is that it requires considerable computational time to select relevant activator-repressor pairs from all possible gene pairs in the search space [31]. Following the classical model, several methods were developed [36] to reduce computational time by utilizing clustering methods to reduce the number of gene pairs. Most of these techniques have improved the performance of the classical model [31] by reducing the required computational time by up to 50% [36, 39]. In addition, fuzzy logic-based tools for GRN modelling, such as FCM requires the use of model learning algorithms such as evolutionary algorithms, which are highly computationally expensive [49]. The time complexity of CS-FCM is $O(n^3MI)$ as reported in [34], where $n$ is the number of nodes (genes) in FCM, $M$ is the number of iterations for data sequences, and $I$ is the number of iterations required for CS-FCM for optimal results. Based on the design of LASSO-FCM, and KFCSFCM, each method requires approximately the same computational time as CS-FCM, as demonstrated in experiments [33–35]. Thus, the required time complexity of these considered FCM-based methods can be simplified as $O(n^3MI)$. In [31] the time complexity of the classical model is given as $O(n^3)$ which is $< O(n^3MI)$. Thus, the classical model is computationally faster than the above-considered FCM-based methods, LASSO-FCM, CF-FCM, and KFCSFCM. The time complexities of the CMI2NI, NARROMI, and MICRAT methods are difficult to compare. However, CMI2NI and NARROMI are computationally intensive as demonstrated in [21, 53] respectively and the time complexity of these methods can be varied depending on the problem.

In MICFuzzy, at pre-processing stage, the information-theory-based approach uses MIC which has been demonstrated to be less computationally intensive [25]. Hence the time complexity of the information-theory-based approach is $O(n^2)$, which is the maximum required time for MIC with default settings [54]. The time complexity of the proposed fuzzy model is $O(n_a n_r n)$ where $n_a$ is the number of activators selected and $n_r$ is the number of repressors selected. The upper bound of $n_a = n\text{-}1$ and $n_r = n\text{-}2$, therefore the time complexity of the proposed fuzzy model is $O(n^3)$. Then the overall time complexity of MICFuzzy is $O(n^2 + n^3)$ or $O(n^3)$. In practice, $n_a << n$ and $n_r << n$, therefore $O(n_a n_r n) << O(n^3)$ which indicates that MICFuzzy is more efficient than the classical model. This is one of the contributions of our work: reducing $n_a$ and $n_r$, using a preprocessing technique, to improve the efficiency of GRN inference.

In this section, we further demonstrated the above-mentioned efficiency improvements of MICFuzzy over the classical model with respect to computational time based on our experiments. Using the MIC-based information theory-based approach as a pre-processing method, in MICFuzzy, the required combinatorial computation was significantly reduced compared with that for the classical model in all DREAM3 and DREAM4 inference problems. This reduction led to an improvement in the efficiency of the proposed model by reducing the required computational time (Fig 7). Based on the experiments, this improvement was more than 55% in all five networks in each sub-challenge, except for the DREAM3 10-gene Net4, which was nearly 40% (Fig 7A). In 50-gene network inference, the reduction of combinatorial computation was more than 60% in all five networks, and for Net2 it was 90.4% (Fig 7B) while outperforming other methods in accuracy also in terms of SS_mean (Table 2) by obtaining a high number of true positives. In 100-gene networks, the improvement in the efficiency was significant obtaining more than 84% in all five networks (Fig 7D) while maintaining high

(a)

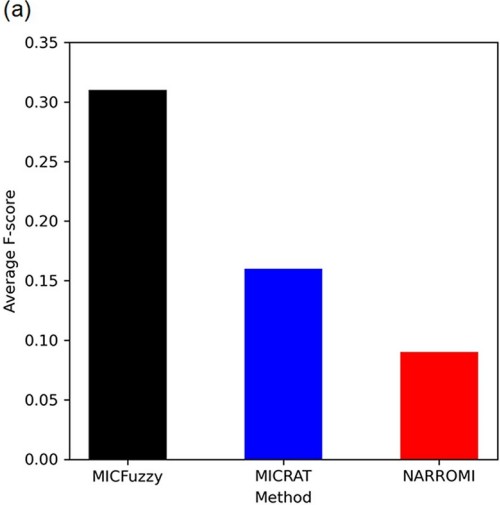

(b)

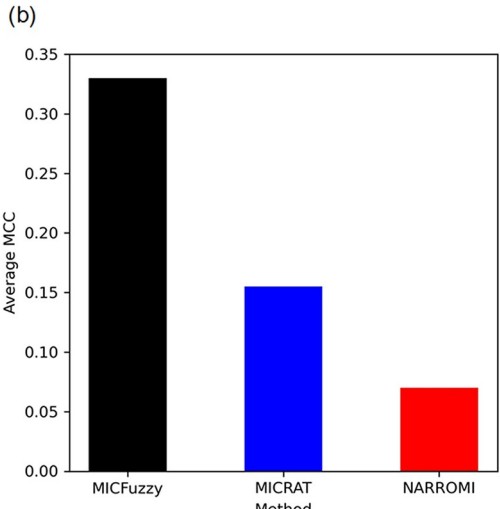

(c)

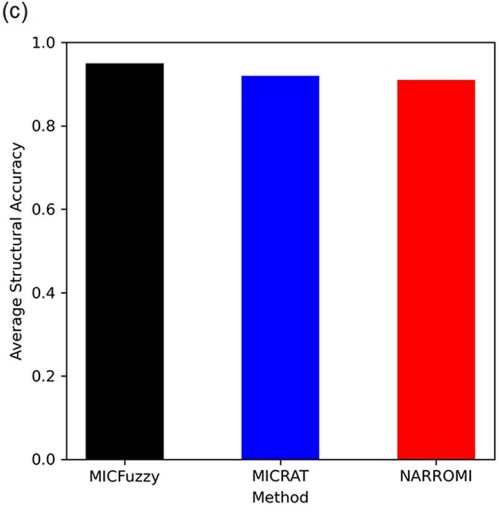

**Fig 6. Comparison of Average F-score, Average MCC and Average Structural Accuracy of MICRAT, NARROMI, and MICFuzzy for inferring DREAM4 100-gene networks.** Results of (a) Average F-score (b) Average MCC and (c) Average Structural Accuracy.

model accuracy (Table 1). Based on these results, it is clear that MICFuzzy is applicable to large-scale network (> 100-gene) inference problems, which will be considered in our future work.

## Experiments with a real-world dataset

The performance of MICFuzzy was tested on an eight-gene network extracted from the SOS DNA repair network in *Escherichia coli*. The exact ground truth for this network is not known, and it is therefore not possible to evaluate the performance of inference algorithms based on the standard performance metrics. Instead, the performance of MICFuzzy was evaluated based on the nine known regulatory interactions reported amongst the eight genes [4, 6, 15] (Fig 8).

On SOS DNA repair network, the performance of the MICFuzzy model was compared with the other well-known state-of-the-art methods MICRAT, GLOBALMIT, PERRIN, MORSHAD and BANJO. For comparison purposes, we followed the same approach as taken in the MICRAT model; the datasets of all four experiments were combined. The MICFuzzy model correctly inferred six regulations, the same number as the MICRAT model. Among the nine known regulations, MICFuzzy identified the regulation type, activation or inhibition of three regulations correctly (Fig 9A) and produced eight false regulations (Fig 9B). MICRAT has no ability to infer regulation types, the interactions between genes which are regulating each other, and has missing information on the number of false regulations identified [15]. Both the MICFuzzy and MICRAT models ignore the presence of self-regulations, but in our future

**Table 2. Comparison of proposed model with other fuzzy models for inferring DREAM3 and DREAM4 networks.**

| Dataset | Method | SS_mean | | | | |
|---|---|---|---|---|---|---|
| | | Net 1 | Net 2 | Net 3 | Net 4 | Net 5 |
| DREAM3 10-Gene | CMI2NI | 0.56 | 0.32 | 0.42 | 0.60 | 0.40 |
| | LASSO-FCM | 0.04 | 0.02 | 0.16 | 0.37 | 0.26 |
| | CF-FCM | 0.51 | 0.48 | 0.61 | 0.47 | 0.44 |
| | KFCSFCM | 0.54 | 0.44 | 0.59 | 0.47 | 0.46 |
| | MICFuzzy | **0.69** | **0.70** | **0.71** | **0.68** | **0.66** |
| DREAM3 50-Gene | CMI2NI | 0.34 | 0.31 | 0.17 | 0.21 | 0.14 |
| | LASSO-FCM | 0.27 | 0.31 | 0.38 | 0.38 | 0.34 |
| | CF-FCM | 0.47 | 0.45 | 0.45 | 0.41 | 0.44 |
| | KFCSFCM | 0.51 | 0.51 | **0.54** | 0.49 | 0.55 |
| | MICFuzzy | **0.55** | **0.52** | **0.54** | **0.55** | **0.60** |
| DREAM4 10-Gene | CMI2NI | 0.54 | 0.31 | 0.48 | 0.54 | 0.64 |
| | LASSO-FCM | 0.07 | 0.11 | 0.09 | 0.15 | 0.04 |
| | CF-FCM | 0.39 | 0.46 | 0.48 | 0.61 | 0.46 |
| | KFCSFCM | 0.51 | 0.53 | 0.60 | 0.76 | 0.59 |
| | MICFuzzy | **0.68** | **0.66** | **0.63** | **0.80** | **0.72** |
| DREAM4 100-Gene | CMI2NI | 0.50 | 0.26 | 0.46 | 0.40 | 0.44 |
| | LASSO-FCM | 0.09 | 0.10 | 0.10 | 0.09 | 0.09 |
| | CF-FCM | 0.69 | 0.52 | 0.56 | 0.61 | 0.57 |
| | KFCSFCM | 0.61 | 0.56 | 0.56 | 0.57 | 0.59 |
| | MICFuzzy | **0.70** | **0.76** | **0.71** | **0.74** | **0.80** |

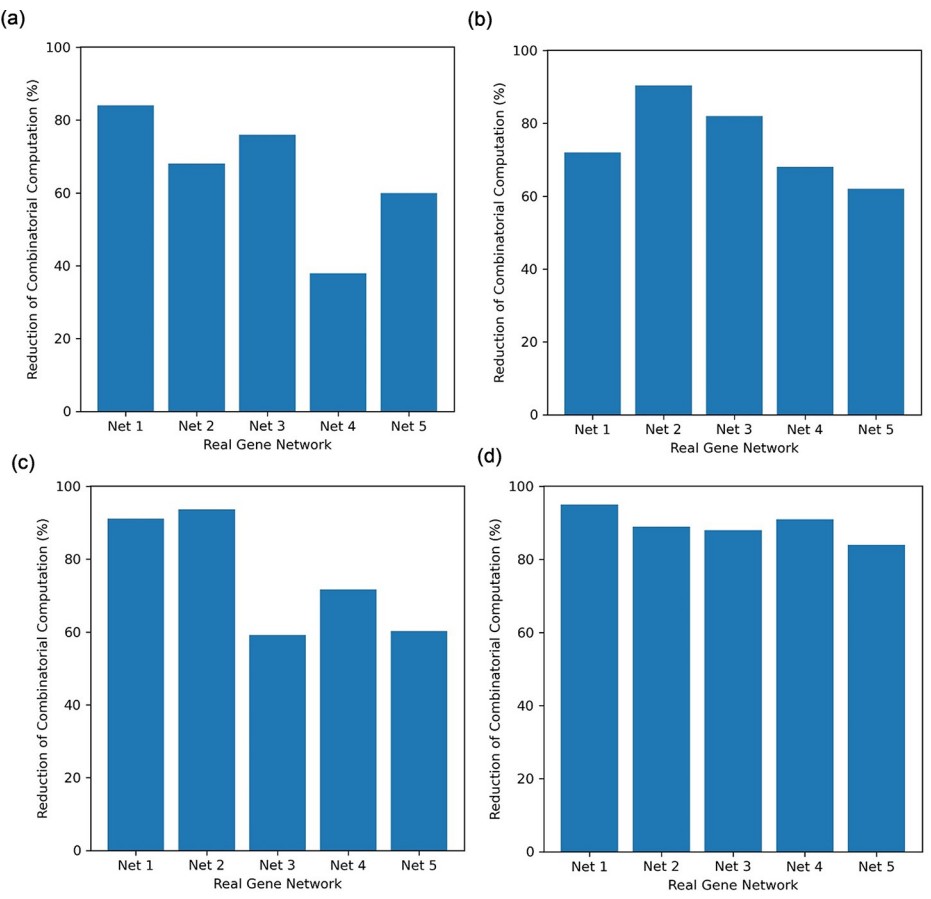

**Fig 7. Reduction of combinatorial computation of the MICFuzzy model compared to the combinatorial computation required for the classical model in inferring DREAM networks.** Inferring results of (a) DREAM3 10-gene, (b) DREAM3 50-gene (c) DREAM4 10-gene and (d) DREAM4 100-gene networks.

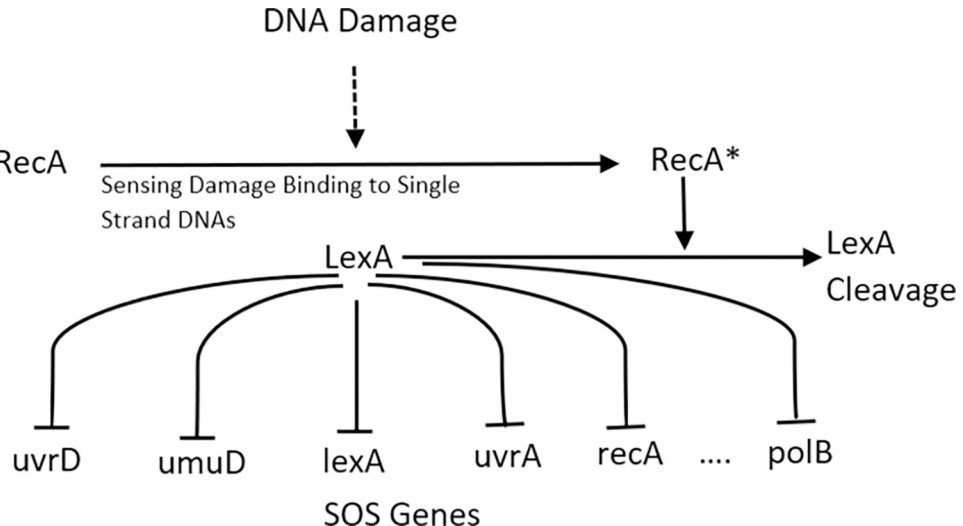

**Fig 8. The bacterial *E. coli* SOS DNA repair target network.**

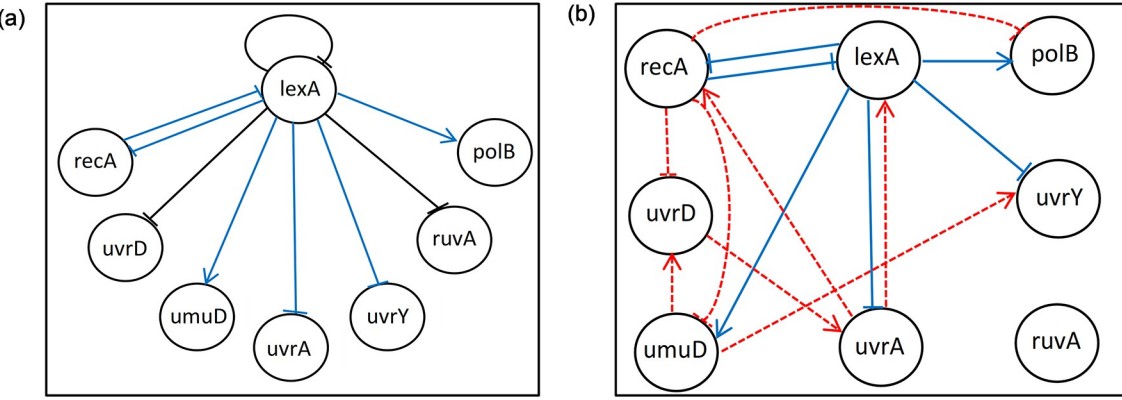

**Fig 9. Structure of SOS DNA repair network reconstructed by MICFuzzy.** (a) Target network with all true regulations. Black lines indicate true regulations which are not predicted by MICFuzzy and blue lines indicate regulations correctly inferred by MICFuzzy with identified regulation type. (b) Both true and false regulations are inferred by MICFuzzy. Blue lines indicate regulations correctly inferred by MICFuzzy and red dashed lines indicate false predictions. In this figure, arrows and barred lines represent positive and negative interactions respectively.

work, self-regulations will be considered. As shown in Table 3, amongst the considered methods, Perrin was the only method that correctly inferred the self-regulation, lexA → lexA. However, by inferring six correct regulations MICFuzzy outperformed Perrin, BANJO, GlobalMIT, and Morshed, which inferred only four, two, five, and four regulations, respectively (Table 3).

Other than the known true regulations, some incorrect regulations were identified by our model, and were justified as novel regulations by reference to the literature [4]. For example, the regulation, uvrA → lexA (Fig 9B) which was inferred by our method was also identified in [4]. Another regulation, uvrA → recA has been reported as a novel regulation [4, 5, 7]. These observations further demonstrate the ability of our proposed model to recover novel regulations while identifying a comparatively high number of known regulations.

## Conclusions

Most of the existing classical fuzzy models and other fuzzy frameworks are limited in their ability to improve the accuracy and efficiency of GRN inference. Our novel hybrid fuzzy

**Table 3. Comparison of true regulations inferred by MICFuzzy with other methods in the SOS DNA repair network.**

| Regulation | Perrin [4] | BANJO [5] | GlobalMIT [7] | Morshed [6] | MICRAT [15] | MICFuzzy |
|---|---|---|---|---|---|---|
| lexA → uvrD | | | | y | y | |
| lexA → lexA | y | | | | | |
| lexA → umuD | | | y | y | y | y |
| lexA → recA | y | | y | y | y | y |
| lexA → uvrA | y | y | y | y | y | y |
| lexA → uvrY | | | y | | y | y |
| lexA → ruvA | | | | | | |
| lexA → polB | | | y | | y | y |
| recA → lexA | y | y | | | | y |

'y'–correctly inferred regulation

model, MICFuzzy provides improved performance and not only infers a large number of true regulatory predictions than existing approaches but also significantly reduces the number of false regulatory predictions. The improvements produced by the proposed hybrid fuzzy model, MICFuzzy are derived from its two main stages, an information theory-based pre-processing stage, and a subsequent fuzzy model, which determines the most suitable regulatory genes from the filtered list obtained from the pre-processing stage. The information theory-based approach is based on a computationally efficient method, MIC, to evaluate gene-gene dependencies, and eliminates irrelevant genes while inferring important regulations. This stage reduces the computational requirements of the fuzzy model. The novel fuzzy model predicts the target gene expression values by considering not only the input gene expression level but also the regulatory relationship strength, which further helps to prioritize highly relevant true regulators while reducing the number of false regulations. Based on the experimental results obtained from the DREAM3 and DREAM4 simulated datasets, along with the real SOS gene dataset, we observed that the information theory-based approach infers a high number of true regulations while the second stage of the fuzzy model has the ability to reduce the number of false predictions while retaining a high number of true regulations, thereby making MIC-Fuzzy superior to other state-of-art methods. The algorithm was applied to microarray gene expression data and can be further enhanced to work with sequencing data (single-cell RNA sequencing) in future developments.

## Supporting information

**S1 Fig. Pseudo-code of MICFuzzy algorithm.**
(PDF)

## Author Contributions

**Conceptualization:** Hasini Nakulugamuwa Gamage, Madhu Chetty, Suryani Lim, Jennifer Hallinan.

**Data curation:** Hasini Nakulugamuwa Gamage.

**Formal analysis:** Hasini Nakulugamuwa Gamage.

**Funding acquisition:** Madhu Chetty.

**Investigation:** Hasini Nakulugamuwa Gamage.

**Methodology:** Hasini Nakulugamuwa Gamage.

**Resources:** Hasini Nakulugamuwa Gamage.

**Software:** Hasini Nakulugamuwa Gamage.

**Supervision:** Madhu Chetty, Suryani Lim, Jennifer Hallinan.

**Validation:** Hasini Nakulugamuwa Gamage.

**Visualization:** Hasini Nakulugamuwa Gamage.

**Writing – original draft:** Hasini Nakulugamuwa Gamage, Madhu Chetty.

**Writing – review & editing:** Suryani Lim, Jennifer Hallinan.

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
