## [Decision Letter · Decision Letter 0]

12 Apr 2023

PONE-D-22-31082MICFuzzy: A maximal information content based fuzzy approach for reconstructing genetic networksPLOS ONE

Dear Dr.Gamage,

Thank you for submitting your manuscript to PLOS ONE. After careful consideration, we feel that it has merit but does not fully meet PLOS ONE’s publication criteria as it currently stands. Therefore, we invite you to submit a revised version of the manuscript that addresses the points raised during the review process.

We look forward to receiving your revised manuscript.

Kind regards,

Prabina Kumar Meher, Ph.D.

Academic Editor

PLOS ONE

Journal Requirements

Additional Editor Comments:

Both the reviewers have raise substantial concerns that need to be addressed carefully. The authors must perform the comparative analysis of the develop approach with the existing methods thoroughly. Please mention the advantage and dis-advantage of the proposed model with respect to existing models. The authors should also provide the source code with step-by-step description for reproducibility of the proposed study.

Reviewers' comments:

Reviewer's Responses to Questions

**Comments to the Author**

1. Is the manuscript technically sound, and do the data support the conclusions?

Reviewer #1: Partly

Reviewer #2: Partly

2. Has the statistical analysis been performed appropriately and rigorously? 

Reviewer #1: No

Reviewer #2: Yes

3. Have the authors made all data underlying the findings in their manuscript fully available?

Reviewer #1: Yes

Reviewer #2: Yes

4. Is the manuscript presented in an intelligible fashion and written in standard English?

Reviewer #1: Yes

Reviewer #2: Yes

5. Review Comments to the Author

Reviewer #1: The paper addresses the interesting topic of inferring gene regulatory network models from gene expression data. In particular, the authors present a method that combines the maximal information coefficient (MIC) to identify regulatory relations with a fuzzy model.

I have some concerns about the implementation of this method, as well as its real contribution concerning existing methods.

Here are some comments

-The paper lacks a description of how the proposed method was implemented. What programming language, libraries, etc., were used. More details about this are needed.

-The estimation of mutual information or MCI is quite challenging when you don't have enough data. A major issue when computing information theory measures like mutual information, and therefore, MIC, is the difficulty of estimating correct values for these measures when the number of time points is limited, a typical problem in gene regulatory network inference problems. What approach did the authors follow to implement MCI? How did they validate that their estimation method was reliable? A better description of this issue is needed.

-The MIC threshold (delta) is computed as the average value. Did the authors try other values? for example, the median, harmonic mean, etc.

-When considering the real application (SOS DNA repair network), it is unclear if the proposed approach is a significant contribution compared to the existing method MICRAT, which has the same preprocessing. Overall, I think more experiments between these two methods are needed to identify the advantages of MICFuzzy over MICRAT better (if they exist).

Reviewer #2: In this study, the authors proposed an improved hybrid method named as MICFuzzy which involves two stages. First, as a pre-processing step, in order to determine the similarity between the target genes and the others, an information theory-based method is provided which uses the maximal information coefficient (MIC) to compute their correlations. By applying this step, the candidate genes with high regulatory relationships are determined and reduces the time complexity of considering the possible genes. In the next step, by applying fuzzy model, from the candidate genes the regulatory genes are nominated for each target gene by inferring the activator repressor gene pairs.

I think the writing of the manuscript is suitable and the method has been evaluated carefully. I have some suggestions which can improve the quality of the manuscript.

1- In the introduction, there are several valuable works which should be reviewed. For example:

a. Turki T, Taguchi YH. SCGRNs: Novel supervised inference of single-cell gene regulatory networks of complex diseases. Computers in biology and medicine. 2020 Mar 1;118:103656.

b. Zhang Y, Chang X, Liu X. Inference of gene regulatory networks using pseudo-time series data. Bioinformatics. 2021 Aug 25;37(16):2423-31.

c. Pirgazi, J., Olyaee, M. H., & Khanteymoori, A. (2021). KFGRNI: A robust method to inference gene regulatory network from time-course gene data based on ensemble Kalman filter. Journal of Bioinformatics and Computational Biology, 19(02), 2150002.

d. Pirgazi J, Khanteymoori AR. A robust gene regulatory network inference method base on Kalman filter and linear regression. PloS one. 2018 Jul 12;13(7):e0200094.

e. Segura-Ortiz A, García-Nieto J, Aldana-Montes JF, Navas-Delgado I. GENECI: A novel evolutionary machine learning consensus-based approach for the inference of gene regulatory networks. Computers in Biology and Medicine. 2023 Mar 1;155:106653.

2- The quality of the figures are not suitable and should be improved.

3- The authors should describe the pseudo code in S1 with more details. Moreover, please discuss about the time complexity.

4- The comparing methods like MICRAT, NARROMI, and … should be cited and explained in brief.

6. PLOS authors have the option to publish the peer review history of their article (what does this mean?). If published, this will include your full peer review and any attached files.

Reviewer #1: No

Reviewer #2: No

---

## [Author Response · Author response to Decision Letter 0]

15 May 2023

We are grateful to the editor and the reviewers for their insightful comments and suggestions. We have incorporated all of the changes suggested by the editor and the reviewers, which have greatly improved the revised manuscript.

Editor’s additional comments:

Both the reviewers have raise substantial concerns that need to be addressed carefully. The authors must perform the comparative analysis of the develop approach with the existing methods thoroughly. Please mention the advantage and dis-advantage of the proposed model with respect to existing models. The authors should also provide the source code with step-by-step description for reproducibility of the proposed study.

Response: Concurring with the editor, we have now revised the “Introduction” Section to include a detailed comparative analysis of the pros and cons of the state-of-the-art methods and our proposed method. The new content will also make it clear how the proposed method addresses key limitations of the earlier work (Pages 2 - 6). These are further demonstrated in the “Results and Discussion” Section based on the results obtained in our experiments (Pages 15 - 21).

The editor has requested that the source code be included. However, instead of source code, we feel it will be more appropriate to include the pseudo-code, which has a language-independent generic form. The readers can easily code in any of their preferred programming languages using the pseudo-code and the step-by-step explanation. This pseudo-code and associated explanation are provided in the supplementary document, S1. However, if source code is needed, it can also be easily made available. 

Reviewers' comments:

Reviewer #1: The paper addresses the interesting topic of inferring gene regulatory network models from gene expression data. In particular, the authors present a method that combines the maximal information coefficient (MIC) to identify regulatory relations with a fuzzy model.

I have some concerns about the implementation of this method, as well as its real contribution concerning existing methods.

Here are some comments.

Query 1: The paper lacks a description of how the proposed method was implemented. What programming language, libraries, etc., were used. More details about this are needed.

Response to Query 1: Thank you for raising this important point. We have included the implementation details of the proposed method, the programming language (Python), and other associated libraries in the sub-section, “Experimental setup and parameter settings” under the “Results and Discussion” Section (Pages 11-12). This additional content is reproduced below for your easy reference. Further, the complete details of the steps to implement the proposed model are explained in the supplementary document, S1.

“MICFuzzy, implemented in Python, uses the minepy Python library [50] to evaluate the MIC between each gene pair in the pre-processing stage. The minepy library has identical performance on biological datasets as its original java implementation (mine.jar) [51] using the default settings published in [25]. We also applied the Scikit-fuzzy library [52], to build the fuzzy control system. MICFuzzy also utilised NumPy, SciPy, pandas, statistics, and Matplotlib scientific libraries. The steps of the implementation of MICFuzzy and its libraries are explained in S1 Fig.”

Query 2: The estimation of mutual information or MCI is quite challenging when you don't have enough data. A major issue when computing information theory measures like mutual information, and therefore, MIC, is the difficulty of estimating correct values for these measures when the number of time points is limited, a typical problem in gene regulatory network inference problems. What approach did the authors follow to implement MCI? How did they validate that their estimation method was reliable? A better description of this issue is needed.

Response to Query 2: We agree with the reviewer’s point. As the reviewer has pointed out, the low sample size of gene expression data is a typical problem for genetic network inference, and it invariably impacts various reconstruction methods including the information theory-based approaches. This is a well-known problem and is widely discussed as the “Curse of Dimensionality” (i.e., small sample size and a large number of variables).

However, since a low sample size invariably affects all known reconstruction methods, in our opinion it should not be associated to consider the effectiveness of the information theory-based approaches. In fact, the effectiveness of information theory-based reverse engineering methods over other measures for GRN reconstruction has been clearly demonstrated, even with small sample sizes. For example, Meyer et al. [1] proposed a mutual information (MI) based approach, minet, and highlighted the suitability of MI due to its ability to deal with several thousands of variables with limited sample size. ARACNE, an information-theoretic approach, showed that applying MI ranking is more robust and outperformed other Bayesian and Relevance network approaches [2]. Another method, MIBNI [3], also use MI to select the initial set of regulatory genes from a small sample size and outperformed several well-known state-of-the-art methods such as CLR [4] and REVEAL [5]. Recent studies, such as CMI2NI [6] and NSCGRN [7], using Conditional Mutual Information (CMI), similar to MI, for quantification of gene associations have demonstrated its effectiveness, in spite of small sample sizes, in GRN inference.

As an enhanced version of MI, apart from its above-mentioned merits, Maximal information coefficient (MIC) also shows impressive performance as a pre-processing approach in GRN inference problems [8,9], especially, by providing a good noise-resistance [9]. In the MICRAT model [10], MIC has been effectively used to identify the initial regulatory network structure. 

Here, we would like to emphasize that MI or MIC, as a pre-processing method, tends to identify a large number of true regulators as top-ranked genes. With additional fine-tuning, the approach can further reduce false positives [11]. Due to this reported success of MIC [10], we decided to implement the MIC method in its original form in our proposed work. We used the minepy Python library [12], based on the original MIC estimator technique [13], to evaluate the MIC between genes. We may add that this library has also been successfully used in a recently developed method, MICTools [14] which is a combination of Total information coefficient (TIC) and MIC to assess the strengths of relationship among variables. It has been demonstrated the potential of MICTools to identify non-functional associations effectively with a relatively low number of samples. 

We agree that the inclusion of key aspects of the above discussion will help to improve the readability of the manuscript, and accordingly, these have now been included in the manuscript. 

Query 3: The MIC threshold (delta) is computed as the average value. Did the authors try other values? for example, the median, harmonic mean, etc.

Response to Query 3: Thank you for this question. As we know, the MIC threshold is a user-defined value and researchers also have a choice of median or harmonic mean in their experiments. In our work, after trialling different options, we observed the ‘average value’ of MIC to be the most suitable and applied it as our preferred option. 

Query 4: When considering the real application (SOS DNA repair network), it is unclear if the proposed approach is a significant contribution compared to the existing method MICRAT, which has the same preprocessing. Overall, I think more experiments between these two methods are needed to identify the advantages of MICFuzzy over MICRAT better (if they exist).

Response to Query 4: In order to address this important suggestion from the reviewer, we have now included additional experimental results of our proposed model on the DREAM4 100-gene dataset. For comparison, we have also included results, reported in [10], of the MICRAT method for this same dataset. Comparison between MICRAT and MICFuzzy shows that MICFuzzy outperforms MICRAT in terms of average Structural Accuracy, and significant improvements can be seen in the average F-Score and average Matthews correlation coefficient (MCC) over MICRAT. 

For real-world datasets, the authors of MICRAT [10] only considered the SOS DNA repair network. Therefore, we executed our method only on the SOS DNA repair network, which allowed us to perform a direct comparison with MICRAT. Since the source code of the MICRAT approach is not publicly available, further comparisons between MICRAT and MICFuzzy using other publicly available real-world datasets were not possible.

This additional analysis is now included in the revised manuscript (Pages 15-16) as follows.

“We evaluated our model based on Average F-score, Average MCC, and Average Structural Accuracy for all five networks in the DREAM4 10-gene and 100-gene datasets, since MICRAT has used these measures for performance evaluation and thus, direct comparison is possible. As shown in Figs 5 and 6, for Average Structural Accuracy, while MICFuzzy showed a slight improvement over MICRAT (Figs 5C and 6C), it showed considerable improvement over NARROMI in 10-gene and 100-gene datasets. Our proposed model outperformed MICRAT and NARROMI substantially in both 10-gene and 100-gene datasets with respect to Average F-score (Figs 5A and 6A) and Average MCC (Figs 5B and 6B) because of its ability to identify a high number of true regulations and reduce the number of false predictions.” 

Reviewer #2: In this study, the authors proposed an improved hybrid method named as MICFuzzy which involves two stages. First, as a pre-processing step, in order to determine the similarity between the target genes and the others, an information theory-based method is provided which uses the maximal information coefficient (MIC) to compute their correlations. By applying this step, the candidate genes with high regulatory relationships are determined and reduces the time complexity of considering the possible genes. In the next step, by applying fuzzy model, from the candidate genes the regulatory genes are nominated for each target gene by inferring the activator repressor gene pairs.

I think the writing of the manuscript is suitable and the method has been evaluated carefully. I have some suggestions which can improve the quality of the manuscript.

Query 1: In the introduction, there are several valuable works which should be reviewed. For example:

a. Turki T, Taguchi YH. SCGRNs: Novel supervised inference of single-cell gene regulatory networks of complex diseases. Computers in biology and medicine. 2020 Mar 1;118:103656.

b. Zhang Y, Chang X, Liu X. Inference of gene regulatory networks using pseudo-time series data. Bioinformatics. 2021 Aug 25;37(16):2423-31.

c. Pirgazi, J., Olyaee, M. H., & Khanteymoori, A. (2021). KFGRNI: A robust method to inference gene regulatory network from time-course gene data based on ensemble Kalman filter. Journal of Bioinformatics and Computational Biology, 19(02), 2150002.

d. Pirgazi J, Khanteymoori AR. A robust gene regulatory network inference method base on Kalman filter and linear regression. PloS one. 2018 Jul 12;13(7):e0200094.

e. Segura-Ortiz A, García-Nieto J, Aldana-Montes JF, Navas-Delgado I. GENECI: A novel evolutionary machine learning consensus-based approach for the inference of gene regulatory networks. Computers in Biology and Medicine. 2023 Mar 1;155:106653.

Response to Query 1: Thank you for this suggested list of valuable work. In the “Introduction” Section, we have now reviewed the work reported in the above references, thereby further improving the quality of the overall manuscript (Pages 3 - 4). 

The additional review included in the “Introduction” Section is reproduced here for easy reference (Pages 3 - 4),

“Gene networks inference using projection and lagged regression (GNIPLR) [14] uses both projection and lagged regression strategies to accurately infer GRNs from time series and non-time series data. LassoCV+RidgeCV [12] is another regression-based model which uses an improved version of both regression methods, incorporating cross-validation to increase model stability and produce accurate results. Both GNIPLR and LassoCV+RidgeCV outperform other existing high-performing regression-based inference methods. However, these regression methods are limited to capturing linear dependencies.”

“KFGRNI is such a method, using Conditional Mutual Information (CMI) based approach to fine-tune a list of genes, selected by the ensemble Kalman filter and regression approach [23]. This method further improves model accuracy by removing false regulations. KFLR [24] uses MI and CMI in the preprocessing stage to eliminate noisy regulations, followed by a Kalman filter-based model averaging approach (a hybrid framework of Bayesian model averaging and linear regression methods) to infer possible regulators. However, both MI and CMI-based inferencing methods cannot discover important non-linear correlations such as sinusoids [15,21]. MI is well suited for use on discrete or categorical data [18], but has known limitations when applied to continuous data, as is found in gene expression datasets.”

Query 2: The quality of the figures are not suitable and should be improved.

Response to Query 2: Thank you for pointing this out. All the figures have been recreated according to the guidelines. The .tif files were checked and generated using the Preflight Analysis and Conversion Engine (PACE) digital diagnostic tool as per the recommendation of Plos One.

Query 3: The authors should describe the pseudo code in S1 with more details. 

Response to Query 3: The pseudo-code and the step-by-step explanation of the pseudo-code are included in the supplementary document, S1.

Query 4: Moreover, please discuss about the time complexity.

Response to Query 4: The time complexity of this method is discussed in detail in the revised manuscript and compared with other state-of-the-art methods as necessary under the sub-section “Impact of pre-processing on computational time” in “Results and Discussion” Section as follows (Pages 17-19),

“The main drawback of the classical model is that it requires considerable computational time to select relevant activator-repressor pairs from all possible gene pairs in the search space [31]. Following the classical model, several methods were developed [36] to reduce computational time by utilizing clustering methods to reduce the number of gene pairs. Most of these techniques have improved the performance of the classical model [31] by reducing the required computational time by up to 50% [36,39]. In addition, fuzzy logic-based tools for GRN modelling, such as FCM requires the use of model learning algorithms such as evolutionary algorithms, which are highly computationally expensive [49]. The time complexity of CS-FCM is O(n3MI) as reported in [34], where n is the number of nodes (genes) in FCM, M is the number of iterations for data sequences, and I is the number of iterations required for CS-FCM for optimal results. Based on the design of LASSO-FCM, and KFCSFCM, each method requires approximately the same computational time as CS-FCM, as demonstrated in experiments [33,34,35]. Thus, the required time complexity of these considered FCM-based methods can be simplified as O(n3MI). In [31] the time complexity of the classical model is given as O(n3) which is < O(n3MI). Thus, the classical model is computationally faster than the above-considered FCM-based methods, LASSO-FCM, CF-FCM, and KFCSFCM. The time complexities of the CMI2NI, NARROMI, and MICRAT methods are difficult to compare. However, CMI2NI and NARROMI are computationally intensive as demonstrated in [53] and [21] respectively and the time complexity of these methods can be varied depending on the problem. 

In MICFuzzy, at pre-processing stage, the information-theory-based approach uses MIC which has been demonstrated to be less computationally intensive [25]. Hence the time complexity of the information-theory-based approach is O(n2), which is the maximum required time for MIC with default settings [54]. The time complexity of the proposed fuzzy model is O(nanrn) where na is the number of activators selected and nr is the number of repressors selected. The upper bound of na = n-1 and nr = n-2, therefore the time complexity of the proposed fuzzy model is O(n3). Then the overall time complexity of MICFuzzy is O(n2 + n3) or O(n3). In practice, na << n and nr << n, therefore O(nanrn) << O(n3) which indicates that MICFuzzy is more efficient than the classical model. This is one of the contributions of our work: reducing na and nr, using a preprocessing technique, to improve the efficiency of GRN inference. 

In this section, we further demonstrated the above-mentioned efficiency improvements of MICFuzzy over the classical model with respect to computational time based on our experiments. Using the MIC-based information theory-based approach as a pre-processing method, in MICFuzzy, the required combinatorial computation was significantly reduced compared with that for the classical model in all DREAM3 and DREAM4 inference problems. This reduction led to an improvement in the efficiency of the proposed model by reducing the required computational time (Fig 7). Based on the experiments, this improvement was more than 55% in all five networks in each sub-challenge, except for the DREAM3 10-gene Net4, which was nearly 40% (Fig 7A). In 50-gene network inference, the reduction of combinatorial computation was more than 60% in all five networks, and for Net2 it was 90.4% (Fig 7B) while outperforming other methods in accuracy also in terms of SS_mean (Table 2) by obtaining a high number of true positives. In 100-gene networks, the improvement in the efficiency was significant obtaining more than 84% in all five networks (Fig 7D) while maintaining high model accuracy (Table 1). Based on these results, it is clear that MICFuzzy is applicable to large-scale network (> 100-gene) inference problems, which will be considered in our future work.”

Query 5: The comparing methods like MICRAT, NARROMI, and … should be cited and explained in brief.

Response to Query 5: We agree that the inclusion of these methods would improve readability. We have now included a brief description of each of these methods in the “Introduction” Section (Pages 2 - 5).

Further, in the revised manuscript, a brief description of each of the Bayesian models used for the performance comparison in inferring E. coli SOS DNA repair network is also included as follows (Page 3),

“Bayesian network modelling is more sophisticated than Boolean modelling and is used to implement high performing reverse engineering approaches [4,5,6,7,8]. Perrin’s method, based on dynamic Bayesian networks is well-suited to the inference of gene regulatory interactions from gene expression data and their derivatives. However, this approach is limited to the inference of small-scale GRNs [4]. Bayesian Network inference with Java Objects (Banjo), is a software package that has been used for both Bayesian and Dynamic Bayesian network inference. Similar to Perrin’s approach, Banjo requires a high computational time, since it uses heuristic search strategies in model learning [5]. Morshed et al. [6] implemented a Bayesian network model to capture both instantaneous and time-delayed interactions that occur concurrently, but the evolutionary search employed requires high computational time [6]. Unlike other Bayesian models, GlobalMIT uses mutual information test (MIT), an information theoretic-based scoring metric as a model learning technique rather than time-consuming local search strategies. However, this approach is poorly scalable since the complex nature of Bayesian modelling makes it computationally intensive when inferring large-scale networks [7].”

As suggested by the reviewer, brief descriptions of NARROMI, CMI2NI, and MICRAT are also included and cited in the “Introduction” Section in the revised manuscript as follows (Pages 4 - 5),

“NARROMI [21] and CMI2NI [22] are other novel MI-based methods for GRN inference. NARROMI, a combination of the ordinary differential equation-based recursive optimization (RO) method and the information theory-based mutual information (MI) method, is an effective method which outperforms most existing methods in terms of accuracy and false positive rates. In this approach, the least relevant regulators for each target are first removed using MI by evaluating pairwise correlations. Then indirect regulators are identified using recursive optimization, which further improved the overall model accuracy [21]. CMI2NI uses a novel association measure, conditional mutual inclusive information (CMI2) which helps to identify direct regulations while eliminating indirect regulations. The main drawback of CMI2 is that its efficiency is negatively impacted by the use of a random method to identify conditional genes [22].”

“Recently MICRAT has been developed, using MIC to infer GRNs as an undirected graph that represents interactions between genes from time series gene expression data. Then the direction of these interactions is determined using a combination of conditional relative average entropy and time course mutual information of pairs of genes [15]. However, these models produce a high number of false regulatory predictions while inferring a high number of true regulations and have no ability to identify the activating or inhibiting effect of regulatory genes.”

References

1. Meyer PE, Lafitte F, Bontempi G. minet: A R/Bioconductor Package for Inferring Large Transcriptional Networks Using Mutual Information. BMC Bioinformatics. 2008 October; 9(461).

2. Adam A Margolin INB, Wiggins C, Stolovitzky G, Favera RD, Califano A. ARACNE: An Algorithm for the Reconstruction of Gene Regulatory Networks in a Mammalian Cellular Context. BMC Bioinformatics. 2006 March; 7(S7).

3. Barman S, Kwon YK. A Novel Mutual Information-based Boolean Network Inference Method from Time-series Gene Expression Data. PLoS ONE. 2017 February; 12(2).

4. Faith JJ, Hayete B, Thaden JT, Mogno I, Wierzbowski J, Cottarel G, et al. Large-Scale Mapping and Validation of Escherichia coli Transcriptional Regulation from a Compendium of Expression Profiles. PLOS Biology. 2007; 5(1).

5. Liang S, Fuhrman S, Somogyi R. Reveal, A General Reverse Engineering Algorithm for Inference of Genetic Network Architectures. Pacific Symposium on Biocomputing. 1998;: 18-29.

6. Zhang X, Zhao J, Hao JK, Zhao XM, Chen L. Conditional Mutual Inclusive Information Enables Accurate Quantification of Associations in Gene Regulatory Networks. Nucleic Acids Research. 2015 March; 43(5).

7. Liu W, Sun X, Yang L, Li K, Yang Y, Fu X. NSCGRN: A Network Structure Control Method for Gene Regulatory Network Inference. Briefings in Bioinformatics. 2022 September; 23(5): bbac156.

8. Yang D, Liu H. Maximal Information Coefficient Applied to Differentially Expressed Genes Identification: A Feasibility Study. Technology and Health Care. 2019; 27(1): 249–262.

9. Liu HM, Yang D, Liu ZF, Hu SZ, Yan SH, He XW. Density Distribution of Gene Expression Profiles and Evaluation of Using Maximal Information Coefficient to Identify Differentially Expressed Genes. PLoS ONE. 2019; 14(7).

10. Yang B, Xu Y, Maxwell A, Koh W, Gong P, Zhang C. MICRAT: A Novel Algorithm for Inferring Gene Regulatory Networks Using Time Series Gene Expression Data. BMC Systems Biology. 2018; 12(115).

11. Akhand MAH, Nandi RN, Amran SM, Murase K. Gene Regulatory Network Inference Using Maximal Information Coefficient. International Journal of Bioscience, Biochemistry and Bioinformatics. 2015; 5(5): 296-310.

12. Albanese D. minepy - Maximal Information-based Nonparametric Exploration. , https://minepy.readthedocs.io/en/latest/; 2013.

13. Albanese D, Filosi M, Visintainer R, Riccadonna S, Jurman G, Furlanello C. minerva and minepy: A C Engine for the MINE Suite and Its R, Python and MATLAB Wrappers. Bioinformatics. 2013; 29(3): 407–408.

14. Albanese D, Riccadonna S, Donati C, Franceschi P. A Practical Tool for Maximal Information Coefficient Analysis. Gigascience. 2018 April; 7(4): 1-8.

---

## [Decision Letter · Decision Letter 1]

22 Jun 2023

MICFuzzy: A maximal information content based fuzzy approach for reconstructing genetic networks

PONE-D-22-31082R1

Dear Dr. Gamage,

We’re pleased to inform you that your manuscript has been judged scientifically suitable for publication and will be formally accepted for publication once it meets all outstanding technical requirements.

Kind regards,

Prabina Kumar Meher, Ph.D.

Academic Editor

PLOS ONE

Reviewers' comments:

Reviewer's Responses to Questions

**Comments to the Author**

1. If the authors have adequately addressed your comments raised in a previous round of review and you feel that this manuscript is now acceptable for publication, you may indicate that here to bypass the “Comments to the Author” section, enter your conflict of interest statement in the “Confidential to Editor” section, and submit your "Accept" recommendation.

Reviewer #1: All comments have been addressed

Reviewer #2: All comments have been addressed

2. Is the manuscript technically sound, and do the data support the conclusions?

Reviewer #1: (No Response)

Reviewer #2: Yes

3. Has the statistical analysis been performed appropriately and rigorously? 

Reviewer #1: (No Response)

Reviewer #2: Yes

4. Have the authors made all data underlying the findings in their manuscript fully available?

Reviewer #1: (No Response)

Reviewer #2: Yes

5. Is the manuscript presented in an intelligible fashion and written in standard English?

Reviewer #1: (No Response)

Reviewer #2: Yes

6. Review Comments to the Author

Reviewer #1: (No Response)

Reviewer #2: The manuscript addresses all of the concerns and suggestions raised during the review process. The authors have made appropriate revisions, and the manuscript is now in an acceptable and publishable state.

7. PLOS authors have the option to publish the peer review history of their article (what does this mean?). If published, this will include your full peer review and any attached files.

Reviewer #1: **Yes: **Gonzalo A. Ruz

Reviewer #2: No

---

## [Editor Report · Acceptance letter]

26 Jun 2023

PONE-D-22-31082R1 

MICFuzzy: A maximal information content based fuzzy approach for reconstructing genetic networks 

Dear Dr. Nakulugamuwa Gamage:

I'm pleased to inform you that your manuscript has been deemed suitable for publication in PLOS ONE. Congratulations! Your manuscript is now with our production department. 

Kind regards, 

on behalf of

Dr. Prabina Kumar Meher 

Academic Editor

PLOS ONE